# Silencing DTX3L Inhibits the Progression of Cervical Carcinoma by Regulating PI3K/AKT/mTOR Signaling Pathway

**DOI:** 10.3390/ijms24010861

**Published:** 2023-01-03

**Authors:** Wei Hu, Yaorui Hu, Yao Pei, Rongrong Li, Fuyi Xu, Xiaodong Chi, Jia Mi, Jonas Bergquist, Lu Lu, Luping Zhang, Chunhua Yang

**Affiliations:** 1School of Chemical Engineering and Technology, Tianjin University, Tianjin 300350, China; 2Shandong Technology Innovation Center of Molecular Targeting and Intelligent Diagnosis and Treatment, School of Pharmacy, Binzhou Medical University, Yantai 264000, China; 3School of Basic Medicine, Binzhou Medical University, Yantai 264000, China; 4Department of Chemistry—BMC, Analytical Chemistry and Neurochemistry, Uppsala University, 75124 Uppsala, Sweden; 5Department of Genetics, Genomics and Informatics, The University of Tennessee Health Science Center, Memphis, TN 38163, USA

**Keywords:** cervical carcinoma (CC), Deltex-3-like, PI3K/AKT/mTOR signaling pathway, therapeutic target, quantitative proteomics

## Abstract

Cervical carcinoma (CC) is the second most prevalent gynecologic cancer in females across the world. To obtain a better understanding of the mechanisms underlying the development of CC, high-resolution label-free mass spectrometry was performed on CC and adjacent normal tissues from eight patients. A total of 2631 proteins were identified, and 46 significant differently expressed proteins (DEPs) were found between CC and normal tissues (*p* < 0.01, fold change >10 or <0.1). Ingenuity pathway analysis revealed that the majority of the proteins were involved in the regulation of eIF4 and p70S6K signaling and mTOR signaling. Among 46 DEPs, Integrinβ6 (ITGB6), PPP1CB, TMPO, PTGES3 (P23) and DTX3L were significantly upregulated, while Desmin (DES) was significantly downregulated in CC tissues compared with the adjacent normal tissues. In in vivo and in vitro experiments, DTX3L knockdown suppressed CC cell proliferation, migration, invasion and xenograft tumorigenesis, and enhanced cell apoptosis. Combination of silencing DTX3L and cisplatin treatment induced higher apoptosis percentage compared to cisplatin treatment alone. Moreover, DTX3L silencing inhibited the PI3K/AKT/mTOR signal pathway. Thus, our results suggested DTX3L could regulate CC progression through the PI3K/AKT/mTOR signal pathway and is potentially a novel biomarker and therapeutic target for CC.

## 1. Introduction

Despite being highly preventable, cervical cancer (CC) is the second most common form of cancer in women living in developing countries [1]. It is estimated that approximately 600,000 women are diagnosed with and more than 300,000 women die from CC worldwide annually [2]. The persistent infection of high-risk (oncogenic) human papillomavirus (HR-HPV), such as HPV16 and HPV18, is responsible for the majority of CC [3]. Furthermore, its occurrence involves viral infection, immune factors, proto-oncogenes and tumor suppressor genes [4]. Currently, most early-stage tumors can be treated with a combination of radiotherapy, chemotherapy and surgery, while patients with advanced malignant tumors have poor prognosis due to tumor metastasis and recurrence [5]. Moreover, cisplatin (DDP), a nonspecific cellular drug that can induce cell death, is the first-line choice in chemotherapy for cervical cancer [6]. The primary or acquired resistance of CC cells to cisplatin is a major challenge in the treatment of cervical cancer. Therefore, the discovery of new molecular targets is urgently needed to treat cervical cancer and improve therapeutic effectiveness of cisplatin to CC.

Proteomics-based technologies, such as nano-liquid chromatography coupled with tandem mass spectrometry (nanoLC-MS/MS), are widely used to identify proteins with complex biological functions in various diseases related to pathogenesis, including human malignancies [7,8,9]. The proteomics approach has been used to identify therapeutic targets for CC from CC cell lines [10,11,12]. Thus, proteomic analysis is a powerful tool to identify the key molecules involved in the development of CC; however, a more comprehensive and in-depth analysis of CC proteomics research is needed. In this study, we present a label-free quantitative proteomics analysis to identify the differentially expressed proteins (DEPs) in CC tissues and adjacent normal tissues from eight patients; we found that Deltex-3-like (DTX3L) was significantly upregulated in CC tissues.

DTX3L is a member of the DTX family, also known as B-lymphoma and BAL-associated protein. The DTX protein has a variety of physiological and pathological roles in cell signaling, growth, differentiation and apoptosis, as well as in the occurrence and progression of various malignancies [13]. Studies have shown that DTX1 promotes the proliferation and aggressiveness of glioblastoma cells through activation of AKT and ERK pathways [14]. DTX3L is highly expressed in multiple tissues, including the thymus, urogenital sinus and rectum [15]. As an E3 ligase, DTX3L has been reported to regulate ESCRT-0 ubiquitination by inhibiting the activity of AIP4 [16]. Furthermore, growing evidence indicates that DTX3L also plays an important role in cancer progression [17,18,19]. In lymphoma, the high expression level of DTX3L contributes to the resistance to DNA-damaging chemotherapeutic agents [20]. However, the expression pattern and function of DTX3L in CC remain unknown.

In our research, we found that the expression level of DTX3L was significantly upregulated in CC tissues compared to that in normal tissues. Furthermore, we investigated the influence of DTX3L on cervical cancer cell biological behavior and cisplatin treatment. Thus, the present study aimed to clarify the expression level and function of DTX3L in CC.

## 2. Results

### 2.1. Protein Identification by Label-Free Quantitative Proteomics

A total of 2631 proteins were identified and quantified from eight pairs of CC and adjacent normal tissue samples, and 708 DEPs were found between CC and adjacent normal tissues (*p* < 0.05). The heat map of DEPs is displayed in Figure 1A using Hiplot (https://hiplot-academic.com, accessed on 1 September 2021). Among them, 45 proteins were significantly upregulated (FoldChange *>* 10, *p* < 0.01) and 1 protein was significantly downregulated (FoldChange < 0.1, *p* < 0.01) in CC tissue samples (Appendix A). Moreover, a volcano plot was generated to highlight the distribution of 46 significant DEPs (Figure 1B). The results of GO analysis of DEPs are shown in Appendix A. DEPs involved with molecular function were enriched in binding and catalytic activity. DEPs linked with biological processes were mainly enriched in the cellular process. DEPs with regards to cellular component were enriched in cellular anatomical entity. Then, we performed Ingenuity Pathway Analysis (IPA) to address the DEPs’ enriched pathways. The results showed that these proteins were significantly involved in elF4 and p70S6K signaling and mTOR signaling pathways (Figure 1C). A PPI network was constructed using the 46 significant DEPs (Figure 1D). It seemed that most key nudes were related with RNA pross, including RNA splicing (SNRPD2, PRPF8), tRNA activation (YARS) or DNA replication (PCNA, NASP). It suggested that cells were active and had strong proliferative capability in CC.

### 2.2. DTX3L Is Overexpressed in CC Tissues

To further confirm protein dysregulation in CC, six proteins with high |log_10_fold change| and low *p*-values were selected, and their expression patterns in CC and adjacent normal tissues were validated by Western blot analysis. As shown in Figure 2A, the results indicated that DTX3L and ITGB6 showed obvious upregulation in all six CC tissues compared with those in adjacent normal tissues. ITGB6 has been reported to be upregulated in CC and affect the progression of CC [21]. However, the expression and function of DTX3L was not reported in CC. Additionally, the TCGA cohort dataset showed that DTX3L expression was significantly higher in both CC tumor tissues and metastatic CC compared with that in normal cervical tissues (Figure 2B,C). Similar results were verified using IHC staining by means of the THPA database (Figure 2D). These were consistent with the proteomic results.

### 2.3. Knockdown of DTX3L Suppresses Proliferation of CC Cells

In order to perform further research on the influence of DTX3L on the proliferation of CC cells, HeLa and SiHa cells were transfected with lentivirus carrying DTX3L shRNA (shDTX3L) or respective control (shCtrl). Knockout efficiency of shDTX3L was confirmed with Western blot and RT-qPCR analysis (Figure 3A,B). A CCK-8 assay showed that the proliferation of HeLa and SiHa cells was significantly suppressed after DTX3L knockdown (Figure 3C). Analogously, the cell-colony-forming assays clearly showed that silencing DTX3L significantly reduced the colony-forming ability of HeLa and SiHa cells (Figure 3D). Together, these data show that DTX3L may promote the proliferation of CC cells.

### 2.4. Knockdown of DTX3L Suppresses Migration and Invasion of CC Cells

We further explored whether DTX3L could regulate the motility and invasion of CC cells. In the wound-healing assay, the DTX3L knockdown group showed significantly reduced ability to repair the gap when compared with that in the control group (Figure 4A). In addition, Transwell matrix invasion assays and migration assays without matrix showed that the number of cells passing through the Matrigel-coated membrane and into the lower chamber in the DTX3L knockdown group was lesser than that in the control group (Figure 4B). Epithelial–mesenchymal transition (EMT) plays an important role in tumor invasion and metastasis. EMT is characterized by loss of epithelial markers such as E-cadherin and upregulation of mesenchymal markers, including N-cadherin and vimentin [22]. Snail could regulate EMT by downregulating E-cadherin during both development and tumor progression [23]. Next, EMT-related protein levels were verified after DTX3L knockdown. According to the results, both SiHa and HeLa cells infected with shDTX3L exhibited a significant reduction in the protein expression level of Snail, Vimentin and N-cadherin, and a significant increase in the protein level of E-cadherin (Figure 4C). Collectively, our findings confirmed that decreased DTX3L expression suppresses the migration and invasion of CC cells.

### 2.5. Knockdown of DTX3L Promotes Apoptosis in CC Cells

The effect of DTX3L on cell apoptosis was analyzed by immunofluorescence, flow cytometry and Western blot analysis. The fluorescence intensity of c-Caspase-3 in the DTX3L knockdown group was significantly higher than that in the control group in both HeLa and SiHa cells (Figure 5A,B). Flow cytometry analysis showed that HeLa and SiHa cells transfected with shDTX3L exhibited higher levels of apoptosis (Figure 5C). Moreover, after DTX3L knockdown, expression levels of the anti-apoptotic protein Bcl-2 and Caspased-3 were reduced, while pro-apoptotic P53 and c-Caspase-3 levels were upregulated in both SiHa and HeLa cells (Figure 5D). These results suggest that the knockdown of DTX3L increased the apoptosis of HeLa and SiHa cells.

### 2.6. Combination of Silencing DTX3L and Cisplatin Treatment Enhances Apoptosis of CC Cells

To probe the combination effect of silencing DTX3L and cisplatin (DPP) treatment, HeLa and SiHa cells transfected with shDTX3L or shCtrl were exposed to various concentrations of cisplatin (0, 2, 4, 8, 16, 32 and 64 μmol/mL) for 24 h. Cell viability was determined by the CCK-8 assay. The IC50 values to cisplatin of the shDTX3L group were lower than those of shCtrl group in both HeLa and SiHa cells (Figure 6A). When treated with the same concentration of cisplatin (0, 5, 10 and 15 μmol/mL), the combination of silencing DTX3L and cisplatin treatment induced higher apoptosis percentage compared to that with cisplatin treatment alone (Figure 6B). Meanwhile, the combination of silencing DTX3L and cisplatin treatment in HeLa and SiHa cells significantly downregulated the expression level of anti-apoptotic protein Bcl-2and increased the expression level of the apoptosis marker c-Caspase-3 (Figure 6C,D).

### 2.7. Knockdown of DTX3L Inhibits PI3K/AKT/mTOR Pathway in CC Cells

IPA analysis yielded DEPs significantly involved in elF4 and p70S6K signaling and mTOR signaling pathways (Figure 1C). Therefore, we verified the levels of proteins related to these signaling pathways in tissues. The expression levels of p-4EBP1/4EBP1, p-p70S6K/p70S6K, p-PI3K/PI3K and p-AKT/AKT were significantly higher in CC tissues than those in adjacent normal tissues (Figure 7A). Furthermore, a Western blot assay revealed that the expression levels of p-4EBP1/4EBP1, p-p70S6K/p70S6K, p-PI3K/PI3K and p-AKT/AKT were significantly reduced after silencing DTX3L in HeLa and SiHa cells (Figure 7B,C). These results suggested that DTX3L could regulate the PI3K/AKT/mTOR pathway in CC cells.

### 2.8. Silencing DTX3L Inhibits Tumorigenesis In Vivo

To further investigate the functional role of DTX3L in tumorigenesis, shCtrl or shDTX3L SiHa cells were subcutaneously injected into athymic nude mice with five mice in each group. Tumor volume was measured every three days from the seventh day after inoculation, and the mice were sacrificed on the 22nd day. As shown in Figure 8A, the shDTX3L tumor grew significantly more slowly than shCtrl tumors did. The final mass of the shDTX3L tumor was also smaller than that of shCtrl tumors (Figure 8B). The knockdown of DTX3L in shDTX3L tumors was determined using Western blot analysis. In addition, the PI3K/AKT/mTOR pathway in shDTX3L tumors was significantly inhibited (Figure 8C). These data suggest that DTX3L knockdown inhibited tumorigenesis in vivo.

## 3. Discussion

Through the application of sensitive and specific proteomic techniques, oncoproteomics has been widely used for investigating novel biomarkers of cancer to help early diagnosis and target therapy. There have been some potential biomarkers identified to play an important role in the development of CC by proteomics, such as High Mobility Group Box 2 (HMGB2) [24], Fibulin 1 (FBLN1) [10] and cyclin-dependent kinase 4 (CDK4) [25]. However, most CC proteomics research is performed with CC cell lines, not CC tissues. In this study, we conducted quantitative proteomics with CC and adjacent normal tissue samples from eight CC patients. A total of 2631 proteins were identified, and 46 proteins showed significant expression differences. Six proteins were selected for further validation, including TMPO, PPP1CB, PP1, PTGES3, ITGB6 and DTX3L. TMPO is related to nuclear proteins expressed in many or all tissues and may be involved in the control of the nuclear structure and cell cycle [26]. PPP1CB is one of the three catalytic subunits of protein phosphatase 1 (PP1) and modulates a variety of cellular functions, including metabolism, cell division and muscle contractility [27]. PTGES3, also known as p23, is an oncogene. p23 has been suggested to be overexpressed in multiple cancers, including breast cancer [28], colorectal cancer [29] and cervical cancer [30]. Desmin is a member of extra-sarcomeric cytoskeletons and plays an important role in muscle contraction [31]. ITGB6 has been found to promote cell proliferation, migration and invasion in CC by activating JAK/STAT signaling pathways [32]. DTX3L shows an altered expression level during tumor progression in multiple carcinomas [13,17,33]. In our results, DTX3L showed obvious upregulation in all six CC tissues compared with that in adjacent normal tissues.

DTX3L is a member of the DTX family, which also contains DTX1, DTX2 and DTX3 [28]. DTX3L was originally identified as a binding partner of BAL1 (PARP9/ARTD9), which is an oncogenic factor in diffuse large B-cell lymphoma (DLBCL) with a prominent immune/inflammatory infiltrate [29]. The overexpression of DTX3L in multiple carcinomas has been investigated. It was found that DTX3L is overexpressed in breast cancer, especially in triple-negative breast cancer, where it functions as a negative regulator of ATRA-induced growth inhibition of breast cancer cells [30]. DTX3L is also highly expressed in gliomas, and its expression level has been shown to correlate with the degree of malignancy and overall prognosis [31]. The overexpression of DTX3L promotes the phosphorylation of STAT1 and represses the transcription of IFN regulatory factor-1 (IRF-1), thus enhancing the proliferation, metastasis and chemoresistance of prostate cancer cells [19]. However, the DTX3L mechanism in CC has not been extensively studied. In our results, we characterized overexpression of DTX3L in CC tissues by leveraging proteomic data and publicly available microarray datasets in TCGA. We also verified upregulation of DTX3L in CC tissues using WB analysis with CC tissue samples.

The deregulation of cell proliferation and the evasion of apoptosis are two hallmarks of cancer cells [32]. We investigated the effect of DTX3L on cell proliferation and apoptosis. Silencing DTX3L reduced cell invasion and migration and increased apoptosis in SiHa and HeLa cells, as confirmed by in vitro experiments. In addition, the results of in vivo assays confirmed the tumor-promotive roles of DTX3L in CC development. Because cisplatin is the first-line chemotherapeutic agent for advanced cervical cancer, we determined the combination effect of silencing and cisplatin treatment. The combination of silencing DTX3L and cisplatin induced a higher apoptosis percentage compared to that with cisplatin treatment alone. Thus, our results showed that DTX3L plays an important role in CC tumorigenesis. To the best of our knowledge, this is the first study to highlight the role of DTX3L in CC.

Moreover, our pathway enrichment analysis demonstrated that DEPs were significantly involved in PI3K/Akt/mTOR signaling. PI3K/Akt/mTOR plays a vital role in basic intracellular functions, also regulating the signaling mechanisms of cell cycle, proliferation, apoptosis and metabolism [34]. PI3K/AKT/mTOR is an important and well-studied intracellular signaling pathway in tumorigenesis [35]. In this study, we found that the expression levels of p-PI3K/PI3K, p-AKT/AKT, p-4EBP1/4EBP1 and p-70S6K/70S6K were significantly higher in CC than those in adjacent normal tissues. However, silencing DTX3L significantly reduced the levels of p-PI3K/PI3K, p-AKT/AKT, p-4EBP1/4EBP1 and p-70S6K/70S6K. Thus, we speculated that the silencing of DTX3L could restrain proliferation, migration and invasion of SiHa and HeLa cells via modulating the PI3K/AKT/mTOR signaling pathway. However, the details of the interaction between DTX3L and the PI3K/AKT/mTOR pathway remain unknown and require verification.

## 4. Materials and Methods

### 4.1. Patient Specimens

Eight pairs of human cervical cancer samples and paired noncancer samples were collected at Shandong Provincial Hospital in accordance with the Declaration of Helsinki. The study was approved by the ethical review board of Binzhou Medical University. All the tissues were independently and histologically diagnosed. All the specimens were stored at −80 °C.

### 4.2. LC-MS/MS Analysis

LC/MS analysis was performed on processed materials. A Q-Exactive Orbitrap mass spectrometer with a nano-electrospray ion source (Thermo Fisher Scientific, Bremen, Germany) was used for all analyses. Dried materials were dissolved in water containing 0.1% formic acid. Peptides were isolated using an EASY-nLC 1200 system and reversed phase liquid chromatography. The utilized approach was a two-step column separation. The analytical column was a 10 cm EASY-column (SC2003, Thermo Fisher Scientific), whereas the pre-column was a 2 cm EASY-column (SC001, Thermo Fisher Scientific). The peptides were eluted at 250 nL/min across a 90 min linear gradient from 4% to 100% ACN. The mass spectrometer was operated in positive ion mode to acquire a survey mass spectrum with a resolving power of 70,000 and a consecutive high-collision dissociation (HCD) fragmentation spectrum of the 10 most abundant ions [36].

### 4.3. Proteomic Data Analysis

The raw proteomics data were analyzed using MaxQuant (version 1.6.5.0) based on the UniProt Homo sapiens database (release 2019-11, with 20199 protein entries). The searching parameters were set as follows: a maximum of 10 and 5 ppm error tolerance for the survey scan and MS/MS analysis, respectively; the specificity enzyme was trypsin; and a maximum of 2 missed cleavage sites were allowed; cysteine carbamidomethylation was set as the static modification; oxidation (M) was set as the dynamic modification. The maximum FDR was set to 1% for peptide and protein identification. Gene ontology and protein class analysis were performed with the PANTHER system (http://pantherdb.org/, accessed on 16 November 2021). Meanwhile, differentially expressed proteins (*p* < 0.05) and their log10-transformed expression ratios were analyzed using Ingenuity^®^ Pathway Analysis (IPA) software (Qiagen, Valencia, CA, USA), and the top canonical pathways associated with the analyzed proteins were listed alongside the *p*-values, which were calculated using right-tailed Fisher’s exact tests. The top 10 pathways were selected for further analysis.

### 4.4. Protein–Protein Interaction (PPI) Network Construction

Varied expression pattern interactions of multiple proteins were searched with the online database Retrieval of Interacting Genes (STRING) (http://string-db.org, accessed on 12 August 2021). Gene lists were provided and searched by selecting “Homo sapiens”. PPI networks were constructed based on the STRING database.

### 4.5. Real-Time Quantitative PCR (RT-PCR)

Total RNA from cultured cells was extracted using TRIzol (Invitrogen, Carlsbad, CA, USA) according to the manufacturer’s instructions. cDNA was synthesized using a high-capacity cDNA reverse transcription kit (TransGen Biotech, Beijing, China) from 2 μg of total RNA. Real-time PCR was performed with a SYBR Green PCR Master Mix (Vazyme Biotech Co., Ltd., Nanjing, China). GAPDH was used as an internal control. Each sample was measured in triplicate. Changes in expression were calculated using the 2^−ΔΔCt^ method [37]. The relative expression ratio is presented as fold change. The primer sequences are shown in Table 1

### 4.6. Western Blot

The total protein from cells and tissues was extracted using a radio immunoprecipitation assay (RIPA) buffer (Sigma-Aldrich, Shanghai, China). The concentration of total protein was detected using a BCA Protein Assay kit (Solarbio, Beijing, China). Then, equal amounts of protein (20 μg) were separated with 10% sodium lauryl sulfate-polyacrylamide gels (SDS-PAGE) and quickly moved to polyvinylidene difluoride membranes (Thermo Fisher Scientific, Boston, MA, USA). Nonspecific binding sites were blocked with PBS-Tween-20 and 5% fat-free milk for 1 h at room temperature. Next, the samples were incubated overnight at 4 °C with the primary antibody. DTX3L antibody (#14795; 1:1000), Vimentin antibody (#5741; 1:1000), Snail antibody (#3879; 1:1000), Cleaved caspase-3 (c-Caspase-3) antibody (#8172; 1:1000), Bcl-2 antibody (#3498; 1:1000), P53 antibody (#2527; 1:1000), PI3K (p85) antibody (#4257; 1:1000), P-PI3K (p85 Tyr458) antibody (#17366; 1:1000), AKT antibody (#9272; 1:1000), P-AKT (Ser473) antibody (#4060; 1:1000), p70S6K antibody (#34475; 1:1000), P-p70S6K (Thr421) antibody (#97596; 1:1000), 4EBP1 antibody (#9644; 1:1000), P-4EBP1 (Thr37/46) antibody (#2855; 1:1000) and β-actin antibody (#3700; 1:1000) were from Cell Signaling Technology (Danvers, MA, USA), DES antibody (AF1414; 1:1000) was obtained from Beyotime Biotechnology (Shanghai, China). TMPO antibody (ab5162; 1:1000), P23 antibody (ab92503; 1:1000), PPP1CB (ab53315; 1:1000) and ITGB6 antibody (ab187155; 1:1000) were purchased from Abcam (Cambridge, MA, USA). N-cadherin (sc-8424; 1:1000) and E-cadherin (sc-71008; 1:1000) were purchased from Santa Cruz Biotechnology (Santa Cruz, CA, USA). On day two, membranes were washed three times for 10 min in TBST and then incubated with the appropriate secondary antibodies (Sino Biological Inc., Beijing, China) for 1 h at room temperature. After another three TBST washes, bands were detected with an ECL chemiluminescence reagent (Amersham Pharmacia Biotech, Freiburg, Germany) using an enhanced chemiluminescence system (Clinx, Shanghai, China).

### 4.7. Cell Culture and Transfection

HeLa and SiHa cells were purchased from Procell life Science & Technology Co, Ltd., Wuhan, China. The cells were cultured in Dulbecco’s Modified Eagle’s Medium (DMEM; Invitrogen, Carlsbad, CA, USA) containing 10% fetal bovine serum (Gibco, Carlsbad, CA, USA) and 1% streptomycin (Gibco, Carlsbad, CA, USA) at 37 °C in a 5% CO_2_ incubator. HeLa and SiHa cells were transfected with lentiviruses carrying shCtrl or shDTX3L, respectively, and Recombinant lentivirus and negative control (NC) lentivirus were purchased from Gene Pharma (Gene Pharma, Shanghai, China). The culture medium was changed to fresh medium 8–12 h after infection. For lentivirus transfection, cells were incubated with lentivirus carrying shDTX3L (target sequence 5′-GCACCATTGTGATTACTTA-3′) or shCtrl (target sequence 5′-CCTAAGGTTAAGTCGCCCTCG-3′) at an MOI of 10 for 12 h. The cells were subsequently cultured in normal medium containing 10% FBS and 2 μg/mL Polybrene for an additional 72 h. The efficiency silencing of DTX3L was identified by Western blot and RT-qPCR analysis.

### 4.8. CCK-8 Assays

CCK-8 assays were used to determine cell proliferation ability and half-maximal inhibitory concentration (IC50) value for cisplatin. For cell viability assessments, 2 × 10^3^ cells/well cells were seeded in a 96-well plate, with triplicate wells per group. The viability of cells was examined every day for 5 days. For IC50 assessment, 1 × 10^4^ cells/well were seeded in a 96-well plate. After 24 h, the cells were treated with cisplatin (Solarbio, Beijing, China) (final concentrations: 0, 2, 4, 8, 16, 32 and 64 μmol/L) for another 24 h. Then, the viability of the cells was examined. Cell viability was measured by a Cell Counting Kit-8 (CCK-8; Dojindo, Kumamoto, Japan).

### 4.9. Wound-Healing Assay

A wound-healing assay was used to assess cell migration. Infected HeLa and SiHa cells were seeded into six-well plates at a density of 2 × 10^6^ cells per well and continuously cultured for another 24 h until 90% confluency. The media were removed, and 200 μL tips were used to make a scratch on the bottom of the plate. The well was washed with PBS to remove damaged cells, and new medium was added. The scratch was photographed at 0 h, 24 h and 48 h after the scratch. Scratch closure was measured using ImageJ.

### 4.10. Colony-Formation Assay

Cell proliferation was detected by a colony-formation assay. Infected cells were seeded in a 6-well plate with complete medium for 2 weeks, which was replaced every 3 days. The colonies with more than 50 cells were then fixed with 4% formaldehyde for 15 min, subsequently stained with crystal violet for 30 min and counted under a microscope. Finally, the data obtained from five stochastic fields were used for statistical analysis.

### 4.11. Transwell Assay

For the Transwell assay, 24-well Transwell chambers with and without Matrigel (BD Pharmingen, San Diego, CA, USA) were, respectively, used to detect the migration and invasion ability of HeLa and SiHa cells. A serum-free cell suspension containing 5 × 10^4^ cells was added to the upper chamber (total 200 μL), and 800 μL of DMEM with 10% FBS was added to the lower chamber. After 48 h, the cells remaining in the upper chamber (non-migrated) were removed, and the ones on the bottom chamber were fixed using methanol and stained with crystal violet to visualize the nuclei. The number of cells that had migrated through the polycarbonate membrane was counted under ×20 magnification.

### 4.12. Flow Cytometry Was Used to Analyze Apoptosis

Into a 6-well plate, 1 × 10^6^ infected cells were inoculated. After 24 h of incubation, the cells were treated with cisplatin or PBS for another 24 h. Then, the cells were digested with a digestion solution without ethylene diamine tetraacetic acid (EDTA), washed twice with PBS and resuspended in 500 µL of 1× binding buffer. Next, 5 µL of 7-aminoactinomycin D (7-ADD) solution and 5 µL of Annexin V-PE were added to the cell suspension for 15 min of incubation at room temperature in the dark. An apoptosis assay was performed with a BectonDickinson flow cytometer BD Canto II (San Jose, CA, USA) according to the manufacturer’s instructions.

### 4.13. Immunofluorescence

The infected cells were plated on coverslips. When cell density grew to 50%, the cells were fixed with 4% parafor maldehyde for 10 min at room temperature, then washed with PBS three times for ten minutes each time. The cells were incubated in immune dyeing washing liquid (0.2%Triton X-100 in PBS) for 10 min, rinsed with PBS three times for 10 min every time, and blocked with 5% bovine serum albumin for 1 h in a 37 °C thermostat. Then, the cells were incubated with primary antibodies (c-Caspase-3, CST, #8172, 1:200) at 4 °C overnight and incubated with the appropriate secondary antibodies for 1 h at room temperature. DAPI staining was performed for total nuclei quantification. Finally, the cells were visualized with a Zeiss LSM880 laser confocal microscope (Zeiss, Oberkochen, Germany).

### 4.14. Subcutaneous Xenograft Experiments

Female nude mice (aged 4–6 weeks) were purchased from Jinan Pengyue Laboratory Animal Breeding Co. Ltd. Ten female nude mice were divided into a control (shCtrl) group and a DTX3L knockdown (shDTX3L) group, with five mice in each group. Into the right groin of each mouse, 8 × 10^7^ lentivirus-infected cells carrying shCtrl or shDTX3L were injected subcutaneously. The length (L) and width (W) of tumors were measured from day 7 post-inoculation every 3 days (tumor volume 3.14/6 × L × W × W). All the mice were sacrificed at day 22. The tumors were separated and measured for weight.

### 4.15. Statistics

GraphPad Prism 8.0.1 (San Diego, CA, USA) was used to analyze the data. All the experiments were conducted in triplicate (technical replicates), and all the data are expressed as the mean ± standard deviation (SD). Differences between two groups were determined using Student’s *t*-test. Differences among multiple groups were determined by one-way analysis of variance (ANOVA). A *p*-value less than 0.05 was considered to be statistically significant.

## 5. Conclusions

In this study, we provided quantitative proteomics of CC and adjacent normal tissue samples. We identified 2631 proteins and 46 significant DEPs, including TMPO, PPP1CB, DES, PTGES3, ITGB6 and DTX3L. The upregulation of DTX3L in CC tissues were vindicated with TCGA, THPA database and Western blot analyses. DTX3L silencing suppressed CC cell proliferation, migration, invasion and xenograft tumorigenesis, enhancing cell apoptosis. The combination of silencing DTX3L and cisplatin induced a higher apoptosis percentage compared to that with cisplatin treatment alone. Further studies showed that the tumor-promoting effect of DTX3L was associated with the PI3K/AKT/mTOR signaling pathway. Therefore, our results suggested that DTX3L is involved in the development of cervical cancer and may become a new strategy for cervical cancer treatment.

## Figures and Tables

**Figure 1 ijms-24-00861-f001:**
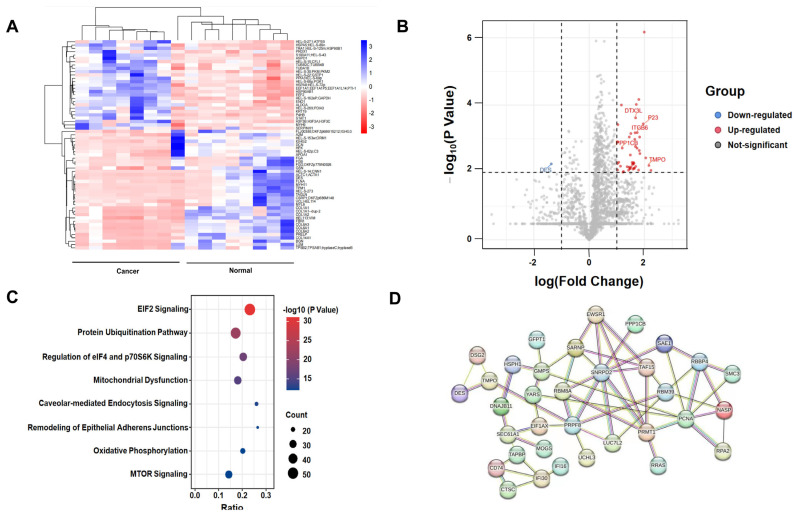
Differential proteomics analysis between CC and adjacent normal tissues. (**A**) Heat map of DEPs is shown between cancer tissues and adjacent normal tissues. The analysis was achieved by using Hiplot (https://hiplot-academic.com, accessed on 1 September 2021). (**B**) Volcano plot illustrating significant DEPs from the quantitative analysis. The −log_10_ (*p*-value) is plotted against log_10_ (Fold Cancer/Normal). The red and blue dots indicate the significantly upregulated and downregulated proteins in CC tissues separately. (**C**) Ingenuity Pathway Analysis (IPA) of 708 DEPs. (**D**) PPI network analysis of 46 significant DEPs.

**Figure 2 ijms-24-00861-f002:**
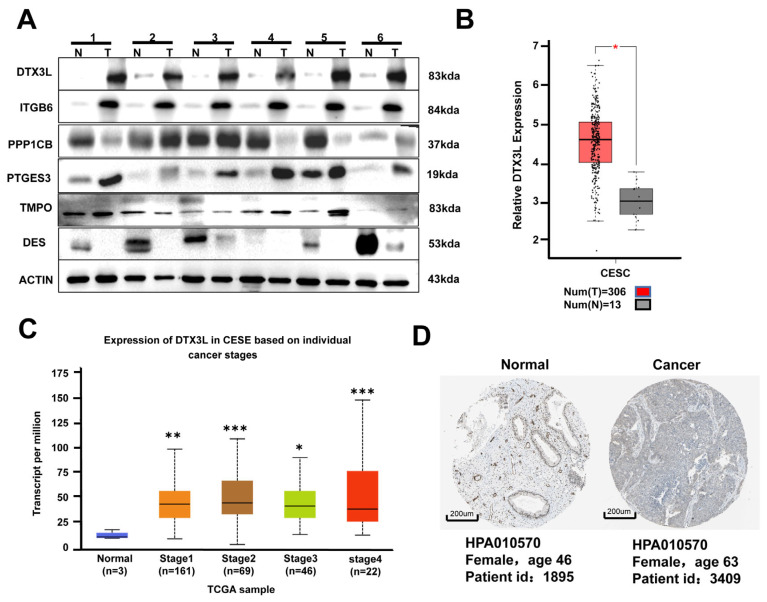
DTX3L was upregulated in CC tissues. (**A**) The expression of six selected proteins in CC and adjacent normal tissues was examined by using Western blot analysis. (**B**,**C**) The expression of DTX3L in CC and normal tissue was analyzed based on the TCGA dataset (Normal vs. Stages, * *p* < 0.05, ** *p* < 0.01, *** *p* < 0.001). (**D**) Representative immunohistochemistry images showing the expression of DTX3L in CC tumor tissues and normal tissues based on the THPA database.

**Figure 3 ijms-24-00861-f003:**
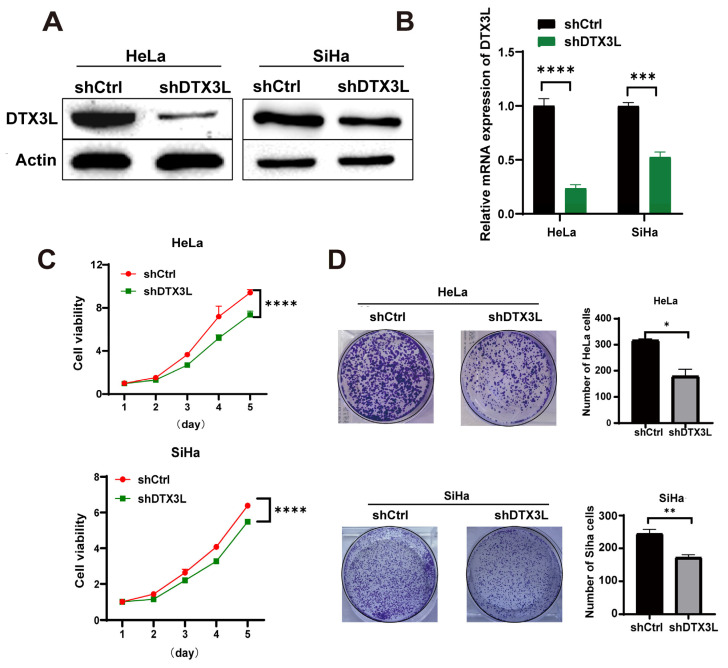
Knockdown of DTX3L suppressed CC cell proliferation. (**A**,**B**) HeLa and SiHa cells were transfected with DTX3L shRNA or control shRNA. The interference efficiencies of shDTX3L were measured by Western blot and RT-qPCR analysis. (**C**) Growth curves of HeLa and SiHa cells were measured by a CCK-8 assay. (**D**) The proliferation of HeLa and SiHa cells was evaluated by colony formation, and the quantified result is shown in the right panel. * *p* < 0.05, ** *p* < 0.01, *** *p* < 0.001, **** *p* < 0.0001.

**Figure 4 ijms-24-00861-f004:**
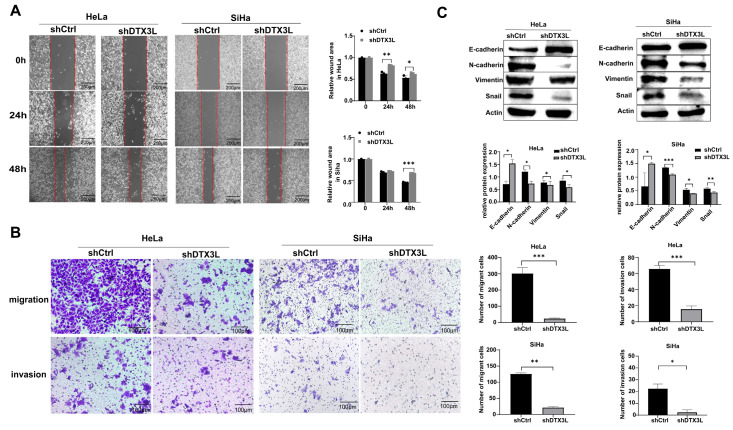
Silencing DTX3L inhibited the migration and invasion of CC cells. (**A**) HeLa and SiHa cells transfected with shDTX3L or shCtrl were photographed 0 h, 24 h and 48 h after scratching the cell surface. Quantification of wound areas were showed in the right panel. (**B**) HeLa and SiHa cell migration and invasion were tested by Transwell assays (crystal violet stain, magnification: 20×). The quantified result is shown in the right panel. (**C**) Expression of EMT-related proteins (E-cadherin, N-cadherin, Snail and Vimentin) in HeLa and SiHa cells transfected with shDTX3L or shCtrl was analyzed using Western Blot analysis. The bar chart in the right shows the ratio of E-cadherin, N-cadherin, Snail and Vimentin to Actin according to densitometry. * *p* < 0.05, ** *p* < 0.01, *** *p* < 0.001.

**Figure 5 ijms-24-00861-f005:**
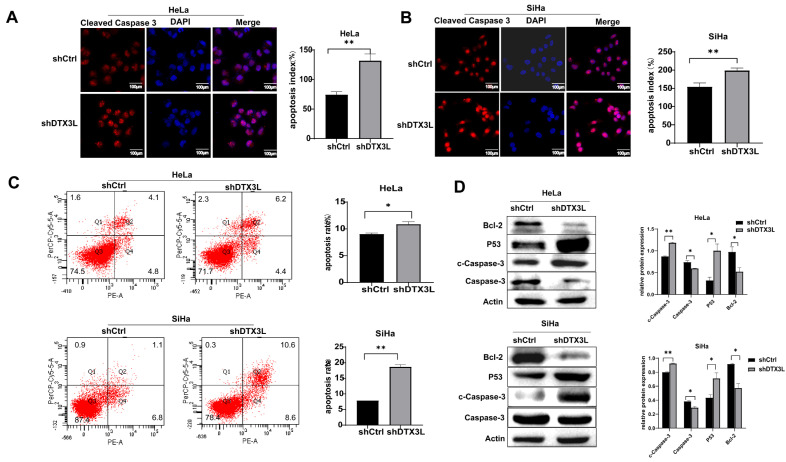
Knockdown of DTX3L promoted HeLa and SiHa cell apoptosis. (**A**,**B**) Immunofluorescence analysis of c-Caspase-3 expression in HeLa and SiHa cells treated with shDTX3L or shCtrl. The quantified result is shown in the right pane. (**C**) The apoptosis ability of HeLa and SiHa cells was detected by flow cytometry, and the percentage of early apoptotic (PE-A+/7-AAD−) and late apoptotic (PE-A+/7-AAD−) cells were combined for analysis. The quantified result is shown in the right panel. (**D**) Protein expression of Bcl-2, P53, Casepased-3 and c-Casepased-3 in HeLa and SiHa cells transfected with shDTX3L or shCtrl were determined by Western blot analysis. The bar chart in the right shows the ratio of Bcl-2, P53, Casepased-3 and c-Casepased-3 to Actin by densitometry. * *p* < 0.05; ** *p* < 0.01.

**Figure 6 ijms-24-00861-f006:**
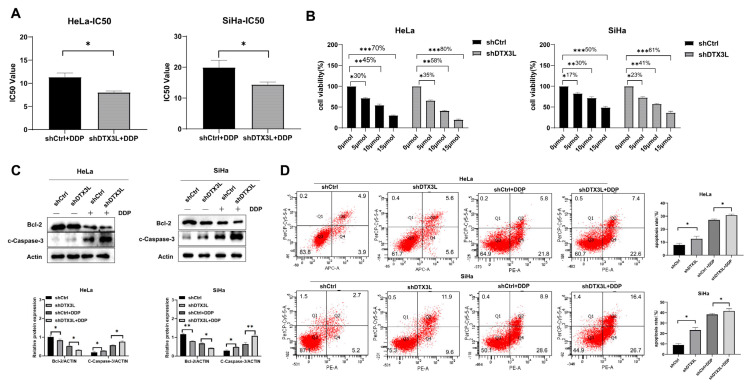
Combination of silencing DTX3L and cisplatin treatment enhanced apoptosis of CC cells (**A**) IC50 values to cisplatin of shDTX3L and shCtrl cells were analyzed by a CCK-8 assay. The used cisplatin concentrations were as follows: 0, 2, 4, 8, 16, 32 and 64 μmol/mL. (**B**) shDTX3L and shCtrl cells were treated with different concentrations of cisplatin. Cell viability was determined by a CCK-8 assay 24 h after treatment. (**C**,**D**) shDTX3L and shCtrl cells were treated with or without cisplatin for 24 h. Protein expression levels of Bcl-2 and c-Casepased-3 were measured by Western blot analysis. Apoptosis ability was detected by flow cytometry, and the percentages of early apoptotic (PE-A+/7-AAD−) and late apoptotic (PE-A+/7-AAD−) cells were combined for analysis. * *p* < 0.05; ** *p* < 0.01; *** *p* < 0.001.

**Figure 7 ijms-24-00861-f007:**
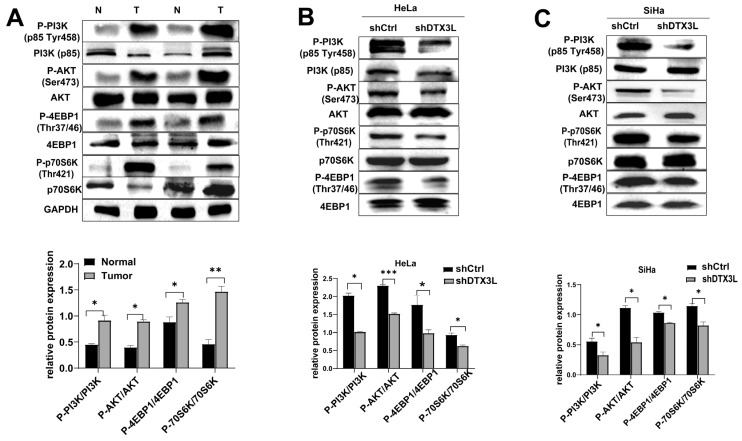
Knockdown of DTX3L inhibited the PI3K/AKT/mTOR signaling pathway. (**A**) The expression levels of PI3K/AKT/mTOR-pathway-related proteins in CC tumor tissues and normal tissues were detected using Western blot analysis. (**B**,**C**) The effect of DTX3L knockdown on the PI3K/AKT/mTOR signal pathway was determined in HeLa or SiHa cells using Western blot analysis. The quantified result is shown below. * *p* < 0.05, ** *p* < 0.01, *** *p* < 0.001.

**Figure 8 ijms-24-00861-f008:**
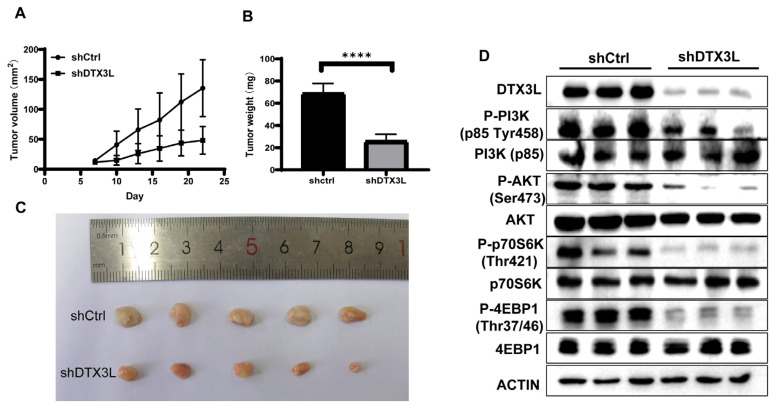
DTX3L knockdown inhibited tumorigenesis in a nude mouse xenograft model. (**A**) Tumor volume was calculated every three days. (**B**) Tumor weight of xenografts on day 22 post-inoculation. (**C**) Representative photographs of the tumors. (**D**) Level of DTX3L and PI3K/AKT/mTOR signaling pathways of representative tumor tissues subjected to Western blot analysis, **** *p* < 0.0001.

**Table 1 ijms-24-00861-t001:** Sequences of primer pairs used for RT-PCR.

Gene	Primer	5′-3′Sequence
DTX3L	F	TGAGTCCTTTGGCACCAT
	R	GGCAAGTATGCAGTTCGC
GAPDH	F	CCCTTCATTGACCTCAACTACATG
	R	TGGGATTTCCATTGATGACAAGC

F: forward; R: reverse.

## Data Availability

The data are contained within the article.

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
