# Peer review of "Silencing DTX3L Inhibits the Progression of Cervical Carcinoma by Regulating PI3K/AKT/mTOR Signaling Pathway"

_ijms, 2023, doi:10.3390/ijms24010861_

Round 1

Reviewer 1 Report

Hu and the colleagues investigated the role of DTX3L in the progression of cervical carcinoma. They observed that silencing the expression of DTX3L suppressed cervical cancer cell proliferation, migration, invasion, and xenograft tumorigenesis, meanwhile enhanced cell apoptosis and cisplatin sensitivity. Their work implied that DTX3L exerted its function by regulating PI3K/AKT/mTOR signaling. Overall, this study is of interest, but there are several comments to address.

Major:

1. When analyzing the results of label-free MS, how did the authors determine the threshold of differentially expressed proteins? Why did they choose this criterion (P < 0.01, Fold change>10 or <0.1)?

2. In Figure 4B, the migration assay of Hela-shCtrl group seemed unconvincing as parts of the chamber was not covered by cells. Please re-check and modify.

3. The morphology of cancer cells in the invasion assay of SiHa cell (shCtrl and shDTX3L) seemed different. Please re-check, clarify and comment.

4. The phosphorylation site of proteins detected by primary antibodies should be shown in Figures For example, the authors should describe the phosphorylation site of PI3K, AKT and 4EBP1 in western blot in Figure7.

5. In Figure 8, the xenografts were very small for observation. When and why were they harvested at this status? Please describe the details and comment.

6. The spelling and grammar should be edited by native English speakers and improved. Some typo erros were shown as follows.

Minor:

1. Line 9 of the Abstract part: “ adjust normal tissues” should be adjacent normal tissues.

2. Line 11 of the Abstract part: slicing should be silencing.

3. Line 12 of the Abstract part: suggest should be suggested.

4. The font size in some pictures are too small to recognize. For example, Figure 1A and 1B.

5. In Figure 2C, the result of statistical comparison was not shown. What kind of statistical method was employed here?

6. The IHC results in Figure 2D was not clear enough for reading and recognizing. Please provide new pictures with higher resolution.

7. The font size of the y-axis in Figure 3C was not consistent.

8. The numbers were covered by the y-axis in Figure 3D. Please redo the picture.

9. The font size of the y-axis in Figure 4A was not consistent.

Reviewer 2 Report

I read with great interest the article " Silencing of DTX3L inhibits the progression of cervical carcinoma by regulating PI3K/AKT/mTOR signaling pathway" by Wei Hu, Yaorui Hu, Yao Pei, Rongrong Li, Fuyi Xu, Xiaodong Chi, Jia Mi, Jonas Bergquist, Lu Lu, Luping Zhang and Chunhua Yang.

In my opinion, the article is well-written, structured and the material is well-chosen.

Results correctly presented and visualized.

Discussion was conducted well, relevant works were cited.

The article presents the current knowledge on the very important topic of cervical carcinoma by regulating PI3K/AKT/mTOR signaling pathway.

Overall, I like the article. Everything is fine. One small note - the authors did not adhere to the IJMS Journal's reference requirements. Please correct.
After revising the references, I believe that the article should be published.

Author Response

Response: We appreciated the reviewer very much. We are very grateful for your approval and positive comments on our manuscript. We have formatted all the references according to the IJMS journal format. Thank you very much for your help.

Reviewer 3 Report

Review on manuscript entitled “Silencing of DTX3L inhibits the progression of cervical carcinoma by regulating 2 PI3K/AKT/mTOR signaling pathway” by Wei Hu et al

In their manuscript, Wei Hu and colleagues performed mass spectrometry on cervical cancer (CC) tissue and adjacent normal tissues from patients to uncover differentially expressed proteins in CC. One of the proteins that were significantly upregulated in CC, DTX3L, was subsequently investigated in a series of in vitro and in vivo assays. The authors propose that DTX3L knockdown suppresses CC cell proliferation, migration, invasion, and xenograft tumorigenesis, while enhancing cell apoptosis and cisplatin sensitivity.

Overall. this is a well-performed study that identifies DTX3L as a novel therapeutic target for cervical cancer and the manuscript (including text, figures and experimental details) is well-curated. Below, I outline several major and minor points that should be addressed by the authors before publication.

Major points

- Results, Section 2.5 and Fig. 5

The authors describe the effects of DTX3L KD with lentiviruses on apoptosis and apoptotic markers. They state that “…pro-apoptotic P53 and c-Caspase-3 levels were upregulated in both SiHa and HeLa cells (Fig. 5D)” (page 6, lines 175-176). The panel in Fig. 5D shows indeed clear increases of P53 after DTX3L KD in SiHA cells (as in Hela cells), but the respective quantification graph bar indicates statistically significant decrease. If this is indeed true, then the above statement should be corrected. In addition, perhaps the authors should give an alternative WB panel that better reflects the WB densitometry data from all their experiments in SiHa cells.

- Results, Section 2.6 and Fig. 6A,B

Here the authors address the effects of DTX3L KD on cisplatin sensitivity of SiHa and HeLa cells and they conclude in the text, section title (as well as abstract and discussion) that “Silencing of DTX3L enhances cisplatin sensitivity of CC cells”. However, I have to admit that I am not really convinced by the IC50 data. The effect shown in Fig6A,B appears not to be significant for both cell lines. For example, IC50 for HeLa cells is decreased by only 6% (and 13% for SiHa cells). These decreases are marginal and, furthermore, there is no statistical analysis to suggest any significance. For statistical significance the authors should have repeated measurements of the IC50 values in separate experiments and then performed the appropriate statistical analysis.

 As such, the main argument that DTX3L KD enhances cisplatin sensitivity of CC cells is not really supported by the IC50 data.  I suggest the authors rephrase this statement to better reflect their findings.

Also, I don’t understand the reasoning in presenting the cell viability data upon different cisplatin concentrations in two different panels, panels A and B. This is the same experimental design but with slightly different concentrations of cisplatin, or not ? It seems to me that just one panel (A or B) should suffice to discuss the differences in cisplatin sensitivity in control and KD cells.

- Results, Section 2.6 and Fig. 6C

The WB panel in Fig 6C shows expression of Bcl-2 and c-Caspase-3 in cisplatin-treated control and KD cells. However, to really compare if there are differences in the cisplatin-induced changes the authors should also incorporate the WBs from untreated cells (e.g. from Fig. 5D) and provide quantification of the cisplatin-induced changes.

Furthermore, in the text (page 7, lines 197-199) the authors state that “Meanwhile, DTX3L knockdown in Hela and SiHa cells inhibited anti-apoptotic protein Bcl-2 expression level, and induced increased apoptosis marker c-Caspase-3 expression level (Fig. 6C)”, without any reference to cisplatin; this omission might be misleading for the reader.  

Minor points

- abstract, page 1, line 28: correct “slicing”

- page 4, line 127: the authors refer correctly to “transduction with lentiviruses” but in other occasions they use the term transfection (e.g. page 7, line 191; page 12, lines 384-385). Unless the latter refer to a shRNA plasmid transfection, please correct.

-page 8, lines 214-215: Please state the specific p-sites for the phospho-specific antibodies that you used, either in the text (Results or Methods) or in the figure legends. For example, p-Akt can be either pS473 or pT308 antibody. Which one was used ?

-page 12, lines 384-385: Concerning control and shDTX3L lentiviruses; were they purchased or constructed? The authors should provide details.

Round 2

Reviewer 1 Report

The authors have provided satisfactory revision and I do no have further comments.